# Fertility-Sparing Surgery versus Radical Hysterectomy in Early Cervical Cancer: A Propensity Score Matching Analysis and Noninferiority Study

**DOI:** 10.3390/jpm12071081

**Published:** 2022-06-30

**Authors:** Antoni Llueca, Maria Victoria Ibañez, Aureli Torne, Antonio Gil-Moreno, Angel Martin-Jimenez, Berta Diaz-Feijoo, Anna Serra, Maria Teresa Climent, Blanca Gil-Ibañez

**Affiliations:** 1Multidisciplinary Unit of Abdominopelvic Oncology Surgery (MUAPOS), Department of Obstetrics and Gynaecology, University General Hospital of Castellon, 12004 Castellón, Spain; serraa@uji.es (A.S.); maclimen@uji.es (M.T.C.); 2Department of Medicine, University Jaume I (UJI), 12004 Castelló de la Plana, Spain; 3Department of Mathematics, University Jaume I (UJI), 12004 Castelló de la Plana, Spain; mibanez@uji.es; 4Unit of Gynaecologic Oncology (ICGON), Endocrinology, Gynaecology and Human Reproduction (IDIBAPS), Hospital Clinic of Barcelona, 08007 Barcelona, Spain; atorne@clinic.cat (A.T.); bdiazfe@clinic.cat (B.D.-F.); 5Gynaecological Oncology Department, Hospital Universitari Vall d’Hebron, Universitat Autonoma de Barcelona, Centro de Investigación Biomedica en Red de Cancer, 08035 Barcelona, Spain; antonioimma@yahoo.es; 6Hospital Universitario Son Llatzer, 07198 Mallorca, Spain; amartin@hsll.es; 7Gynaecological Oncology and Endoscopy Unit, Department of Obstetrics and Gynaecology, University Hospital 12 de Octubre, Research Institute i+12, 28041 Madrid, Spain; blancalabacin@hotmail.com

**Keywords:** trachelectomy, early cervical cancer, fertility preservation treatment, minimally invasive surgery, radical hysterectomy, fertility-sparing surgery

## Abstract

Objective: Fertility-sparing surgery (FSS) is the treatment of choice for patients with early cervical cancer (ECC) and fertility desire, but survival rates compared to radical hysterectomy (RH) have been scarcely reported. The aim of this study was to analyse the oncological outcomes of FSS compared to a balanced group of standard RH. Methods: A retrospective multicentre study of ECC patients who underwent FSS or RH was carried out in 12 tertiary hospitals in Spain between January 2005 and January 2019. The experimental group included patients who underwent a simple and radical trachelectomy, and the control group included patients who underwent RH. Optimal 1:1 propensity score (PS) matching analysis was performed to balance the series. Results: The study included 222 patients with ECC; 111 (50%) were treated with FSS, and 111 (50%) were treated with RH. After PS matching, a total of 38 patients in the FSS group and 38 patients in the RH group were analysed. In both groups, the overall survival (HR 2.5; CI 0.89, 7.41) and recurrence rates (28.9% in the FSS group vs. 13.2% in RH group) were similar. The rate of disease-free survival at 5 years was 68.99% in the FSS group and 88.01% in the RH group (difference of −19.02 percentage points; 95% CI −32.08 to −5.96 for noninferiority). In the univariate analysis, only tumour size reached statistical significance. Conclusion: FSS offers excellent disease-free and overall survival in women with ECC with fertility desire and is not inferior compared to RH.

## 1. Introduction

In Spain, both the incidence and the mortality of cervical cancer are lower than in neighbouring countries, at 6.8 and 2.8 cases per 100,000 women per year. Current surgical treatment of cervical cancer is limited to the early stages of the disease, and radical hysterectomy is the gold standard for cervical cancer treatment in these stages [1,2,3].

This technique provides an adequate curative treatment of the disease but in turn has a negative impact on future fertility in young patients. Novak, in 1952, and Aburel, in 1957, described a laparotomic radical trachelectomy technique to preserve the fertility of young patients with early cervical cancer. At that time, the technical difficulty of performing this technique made it unpopular [4,5].

The trachelectomy technique became more popular in the early 2000s, as described by Dargent [6], who revived the radical trachelectomy technique but used a vaginal approach with minimal invasiveness for lymphadenectomy.

Different surgical approaches to early-stage uterine cancer can produce different results in terms of disease-free and overall survival, as already demonstrated in the LACC (Laparoscopic Approach to Cervical Cancer) trial [7]. Although this multicentre randomised trial was focused on radical hysterectomy, it also raises a series of questions about the extrapolation of these results to any type of surgical intervention or approach for the treatment of cervical cancer. The IRTA (International Radical Trachelectomy Assessment) study [8] concluded that, for small tumours, the results and preservation of fertility are comparable regardless of the approach. Until now, in the absence of prospective randomised studies, the recommendations regarding fertility-sparing surgeries have been based on observational studies and the decisions of tumour committees where the experience of the surgeon has prevailed.

The primary objective of this study was a balanced comparison between the classical radical treatment technique and fertility-preserving surgery (FSS) in the early stages of cervical cancer in terms of oncological outcomes.

A secondary objective was to evaluate the hypothesis that FSS is not inferior to radical hysterectomy with respect to the percentage of patients who are disease-free at 5 and 10 years after surgery.

## 2. Materials and Methods

### 2.1. Study Design

A multicentre, retrospective cohort study of early cervical cancer patients who underwent FSS or radical hysterectomy was carried out in 12 tertiary hospitals in Spain between January 2005 and January 2019. Eligible patients had squamous carcinoma, adenocarcinoma, or adenosquamous carcinoma, stages IA1 with lymphovascular space invasion, IA2, and IB1 (<2 cm and 2–4 cm) (FIGO 2009) [9], and no evidence of lymph node involvement and/or other metastasis in the preoperative evaluation. Exclusion criteria included aggressive histologies (different from the ones described above), and where intraoperative frozen section analysis confirmed positive lymph nodes or peritoneal spread.

### 2.2. Methods

FSS included simple trachelectomy and radical trachelectomy. The surgical approach (laparotomy, laparoscopy, robot-assisted laparoscopy, or vaginal hysterectomy), level of radicality, and the decision of whether to include sentinel lymph node mapping were based on surgeon and centre protocols.

Different types of radical hysterectomy were performed in patients with no desire to preserve fertility using different surgical approaches [1,2].

Surgical outcomes were measured on the basis of surgical reports. Intraoperative complications were recorded. Postoperative complications were considered during the first 3 months following the Clavien–Dindo classification [10].

The use of adjuvant chemotherapy and/or radiation was at the discretion of the tumour boards of the different centres. Progression-free survival (PFS) was defined as the time from surgery to diagnosis of local recurrence or metastasis. Overall survival (OS) was defined as the time from surgery to the date of death or last follow-up. The appearance of cervical, paracervical, or vaginal lesions was considered local recurrence. Events involving the adnexa were considered pelvic recurrences; when the disease affected the lymph nodes, it was considered positive lymph node disease, and, when recurrences appeared in distant organs, they were considered metastases. This study was approved by the Clinical Research Ethics Committee of the Hospital Clinic of Barcelona (study protocol 87/2019) as the reference centre and by the Institutional Review Boards of the participating hospitals.

### 2.3. Statistical Analysis

The variables were summarised according to their nature using means and standard deviations or frequencies and percentages. Similarly, univariate analysis was performed to analyse possible differences in the quantitative covariates of the patients treated with both surgeries. Parametric (*t*-test) or nonparametric tests (Mann–Whitney test) were applied depending on whether the analysed variable followed or did not follow a Gaussian distribution. Chi-square or Fisher’s exact tests were used for the qualitative variables. Given the low number of deaths, we did not focus our analysis on overall survival but on disease-free time, considering recurrence as the event to analyse. From now on, when talking about survival, we mean disease-free survival.

Optimal 1:1 propensity score (PS) matching [11] analysis was performed to balance the series of patients who underwent radical hysterectomy or FSS, making them comparable regarding the set of covariates of interest in the study. The analysis was performed to obtain a subsample of patients verifying that covariates were balanced across both surgical groups. This score matching method is optimal in the sense that the sum of the absolute pairwise distances in the matched sample is as small as possible and provides the best match among the methods tested. Neither the optimal propensity score method nor any of the other matching methods tested preserved all the patients with recurrences in the balanced subsample. This is a problem for the posterior survival analysis, where it is important to have as many recurrence cases as possible. As a first step of PS matching, we found those patients who underwent radical hysterectomy that best matched the patients who underwent preservation surgery and suffered a recurrence. In the second step, we obtained the patients who underwent preservation surgery that best matched the patients who underwent radical hysterectomy and suffered a recurrence. Then, a third optimal 1:1 PS matching analysis was performed with the patients who were unmatched in the two previous steps. For this purpose, radical hysterectomy was considered the control arm, and FSS was considered the experimental arm. Finally, the three subsets were merged into a single matched sample. Absolute standardised mean differences <0.10 were desired in all comparisons.

A noninferiority study was conducted to compare the oncological outcomes of both surgical treatments. Kaplan–Meier survival curves were compared, and a univariate Cox model was used to estimate the hazard ratios (HRs) and their 95% confidence intervals for the effect of the different surgeries on disease-free survival. In this case, the noninferiority of FSS with respect to hysterectomy was concluded using Wald’s confidence interval using the estimated confidence limits of the HR. If the upper confidence limit of the HR was less than the noninferiority margin (1.1), then noninferiority could be assessed. The assumption of proportional hazards was tested and assessed for all analyses that involved hazard ratios [12].

To check if any of the factors considered in the study were risk factors for the disease-free survival time of a patient, univariate and multivariate Cox regression models were fitted.

To compare the effect of the factors considered in the study on the disease-free survival time in both surgical groups, HRs and their 95% CIs were calculated.

To analyse the noninferiority of FSS with respect to hysterectomy in disease-free survival rates at 2.5 and 5 years of follow-up, predictions based on Kaplan–Meier curves and confidence intervals for the difference in survival rates were calculated. In previous studies involving other types of cancer, noninferiority margins of 6–8 percentage points (absolute value) were considered to be clinically acceptable [13,14,15]. As described before [7], we considered a margin of 7.2 percentage points. Therefore, if the upper confidence limit of the confidence interval of the difference in the disease recurrence rates was less than the noninferiority margin (−7.2), then noninferiority could be rejected. Therefore, the noninferiority of FSS with respect to hysterectomy was assessed if the lower boundary of the 95% confidence interval of the difference in survival rates was greater than −7.2 percentage points.

Statistical analysis was performed using R (R Core Team, 2021, Vienna, Austria. Available at: https://www.R-project.org/, accessed on 1 May 2022) [16]. The “MatchIt” [17] and “cobalt” [18] packages were used to perform and visualise the results of the PS matching analysis, and the “survival” [19], “EquiSurv” [20] and “survminer” [21] packages were used for the survival analysis. Unless otherwise stated, all analyses were performed with a two-sided significance level of 0.05.

## 3. Results

The study included 222 patients with early cervical cancer; 111 (50%) were treated with radical hysterectomy, and 111 (50%) were treated with fertility-sparing surgery.

### 3.1. Unmatched Series

The clinicopathological characteristics of the total unmatched series are shown in Table 1.

Regarding the surgical approach, the majority of FSSs were performed vaginally (64.9%), and the minimally invasive approach was predominant in radical hysterectomies (63.1%) (*p* < 0.001). During FSS, intrasurgical complications occurred in 5.4% of the patients (five bladder injuries and a uterine artery tear) (six patients vs. three patients in conventional surgery *p* = 0.48). Seven and zero patients suffered major postoperative complications (Clavien–Dindo grade III–IV) after conventional surgery and FSS, respectively (*p* = 0.001). In the analysis of recurrences, 16 patients had recurrence of their disease: 11 patients after fertility-preserving surgery (recurrence rate 9.9%) and five patients after conventional surgery (recurrence rate 4.5%). Disease-free survival in patients after FSS was 15 months (95% CI 8.5–21.5) vs. 19 months for conventional surgery (95% CI 3.9–34), and the differences were not statistically significant.

The global mean follow-up time was 62.8 months, with a standard deviation (SD) of 45, while it was 59.6 (SD 43.3) and 66 (SD 46.6) months for radical hysterectomy and FSS, respectively. No significant differences were observed.

### 3.2. Recurrence Patients

The mean time until recurrence of the patients who suffered a recurrence was 23.10 (SD 20.38) months. The minimum recurrence time was 1.63 months, and the maximum time was 64.7 months.

The mean time to recurrence for patients with recurrence after hysterectomy was 26.75 (SD 24.54) months, whereas, for patients with recurrence after trachelectomy, it was 21.45 months (SD 19.29) (*p* = 0.6837).

The clinicopathological characteristics of the patients with recurrence are described in Table 2.

Regarding tumour size, the percentage of patients with recurrence in lesions larger than 2 cm was significantly higher than that of recurrence in lesions smaller than 2 cm (*p* = 0.0083), which was also demonstrated in the FIGO classification, with a higher percentage of recurrences in patients with FIGO 2009 stage IB1 >2 cm (*p* = 0.0343).

Of the 109 operations performed laparoscopically, 10 (9.2% of the total) had recurrences compared to none of the laparotomic operations, and six (5.5%) of the operations were performed vaginally. Among the patients who suffered recurrence, there was no significant relationship between the approach route and the type of intervention (*p* = 0.5834).

### 3.3. Matched Series

As seen in Table 1, one of the main differences between the control and treatment groups was age. Figure 1 shows the distribution of the variable age at diagnosis in both treatment groups. At ages greater than 40 years, hysterectomy was predominant over trachelectomy, and from the age of 45 in the sample, we found only one patient treated with trachelectomy (55 years, 0.9% of the sample), compared to 63 patients treated with hysterectomy (56.76% of the sample). As seen in Table 2, this difference also remained between the patients with recurrence. As this variable was not balanced in the two surgical groups, to reduce the bias due to this possible confounding variable, we proposed a subsample with homogeneous groups before moving onto the survival study. When balancing by age, we would lose a large sample size, since we would be left mainly with women under 45 years of age, of which we had 110 in the trachelectomy group but only 48 in the hysterectomy group. Thus, looking for an age-balanced sample with the same sample size in the two surgical groups, we would be left with a maximum of 48 + 48 = 96 patients, which could now be unbalanced according to the rest of the covariates.

Now, although we only found differences in both groups in the variables age and approach route (Table 1), we also considered the covariates FIGO classification 2009, histology, and tumour size in the propensity score to ensure that the resulting sample was also balanced with respect to these variables. After the propensity score, a balanced sample of 76 patients was obtained. Half of them were treated with the classical radical treatment technique, and the other 38 patients were treated with fertility-preserving surgery. A Love plot, as shown in Figure 2, was a clean way to visually summarise balance, showing the mean difference of the distribution of each variable in both surgical groups on the initial dataset (all) and the matched subsample (matched). As shown in Table 1, this plot again showed that balance was quite poor prior to matching the variables age, surgical approach, and Figo Classification 2009, but the matching improved the balance across all covariates, as most were within a threshold of 0.1.

For a clearest overview of the matched subsample, the clinicopathological characteristics of the included patients are described in Table 3.

All patients were negative for intraoperative nodes and, therefore, underwent cervical surgery. In one patient with a tumour size of 2–4 cm and FSS, micrometastasis was diagnosed at the time of ultrastaging. She underwent a hysterectomy and received adjuvant therapy.

### 3.4. Survival Analysis

Continuing with the matched sample, Figure 3 shows the Kaplan–Meier curves to compare the probability of disease-free survival of both surgical procedures.

We compared the null hypothesis that stated that there was no difference between the two survival functions with the alternative that there was (logrank test), obtaining a *p*-value of 0.07. Therefore, strictly speaking, we did not detect significant differences between the functions of disease-free survival of radical hysterectomies and FSS.

Additionally, to investigate the association between the disease-free survival time of patients and the two surgical procedures, Cox proportional hazards models were performed, obtaining an HR associated with fertility-sparing surgery against hysterectomy of 2.5 (CI 0.89, 7.41). With this confidence interval, noninferiority of FSS could not be assessed.

To check if, in addition to the surgical procedure, any of the factors considered in the study were risk factors for the disease-free survival time of a patient, univariate and multivariate Cox regression models were conducted. Just one of the factors considered in the study was significantly associated with survival in a univariate analysis, and none of them reached statistical significance in the multivariate analysis (Table 4). Due to the clear collinearity among the covariates “tumour size” and “Figo Classification”, this last variable was not included in the Cox regression model.

To compare the effect of the factors considered in this study on the disease-free survival time in both surgical groups, HRs and their 95% CIs can be seen in Table 5.

In none of these cases could the noninferiority of FSS over hysterectomy be concluded.

Table 6 evaluates the differences in disease-free survival rates at 2.5 and 5 years after surgery. The rate of disease-free survival at 5 years was 68.99% for fertility-sparing surgery and 88.01% for radical hysterectomy (difference −19.02 percentage points; 95% CI, −32.08 to −5.96 for noninferiority). The lower margin of this interval included the noninferiority margin of −7.2 percentage points; thus, noninferiority could not be confirmed. To affirm noninferiority, the entire confidence interval should be above −7.2 percentage points. Figure 4 shows the CI obtained for the difference in disease-free recurrence rates at 5 years.

## 4. Discussion

### Principal Findings

In this multicentre retrospective study, we found no difference in terms of disease-free rates at 5 years after surgery between FSS and radical hysterectomy, even after adjusting for potential confounding variables due to unbalanced groups (*p* = 0.07). In the study of the noninferiority of FSS (experimental group) versus hysterectomy (control group), we could not affirm that FSS is inferior to radical hysterectomy in the disease-free survival of early cervical cancer; however, as we can see in Figure 4, FSS could be up to 23 percentage points lower than the classic treatment.

In 2018, Tseng et al. [22] evaluated 2717 patients with cervical cancer in FIGO 2009 1B1 stages from the SEER database (Surveillance, Epidemiology, and End Results). All patients underwent lymphadenectomy, but only 125 women had preserved fertility and underwent conisation/trachelectomy, while the remaining 2592 women underwent hysterectomy. There were no differences in disease-specific survival at 10 years between the two cohorts. This is consistent with our results.

Despite the rising trend of delaying pregnancy in a person’s early 30s in Spain, fertility preservation surgery is not very often offered after a diagnosis of early cervical cancer [23].

Although fertility-sparing surgery for cervical cancer should exclusively be undertaken in specific gynaecologic oncology centres with highly trained surgeons, every young woman with a desire to preserve fertility should be counselled about the possibility of fertility-sparing surgery and be referred to a tertiary care hospital with comprehensive expertise in this kind of oncologic therapy [24].

Nevertheless, the oncological outcomes of Spanish women with cervical cancer and a tumour size <2 cm are consistent and in concordance with those published in the literature [25].

In our study, the only independent factor that impacted the oncological outcomes after fertility-sparing surgery was tumour size. Patients who underwent trachelectomy with a tumour size of 2–4 cm had an almost sixfold increased risk of recurrence compared to those with tumours <2 cm, even with a higher rate of adjuvant treatment. Publications focusing on oncological outcomes in tumours larger than 2 cm are heterogeneous. The reported recurrence rates for patients with tumours larger than 2 cm vary tremendously, ranging from 0% to 38% [15].

An essential criterion for performing FSS is that the nodes are negative. Park et al. [26] suggested that lymph node assessment with sentinel lymph node biopsy and/or full lymph node dissection should be performed in this population. Although some authors [27] advocate for the evaluation of the nodes after the administration of neoadjuvant chemotherapy, considering that neoadjuvant chemotherapy can eliminate the possibility of micrometastases and, thus, is able to preserve fertility in most cases, all the patients in this series were negative for intraoperative nodes and, therefore, underwent surgery. Most of the teams represented in the study considered that patients with lymph node dissemination should be treated with adjuvant chemoradiotherapy and, therefore, have no surgical indication. This is in accordance with current guidelines from the National Comprehensive Cancer Network and the European Society for Medical Oncology [3,28].

Tumour size (and, thus, disease stage) is the second main criterion when considering the indication and the type of fertility-sparing surgery. In our data, tumours >2 cm recurred in 14.8% of cases compared to those <2 cm, which only recurred in 3.9% of cases (*p* < 0.0083) for both classical and experimental techniques. This was demonstrated with an HR value of 2.09 for tumours >2 cm. This is in agreement with other authors who affirmed the same [26] This adverse result observed in our series is independent of histology, lymphovascular space invasion, previous conisation, or surgical approach.

Regarding the surgical approach, we found no differences when comparing vaginal or minimal invasion routes, contrary to what was already published in the LACC trial [7]. It is noteworthy that in our matched series, 60.5% and 65.8% of FSSs and radical hysterectomies, respectively, were performed with a minimally invasive approach. However, a tendency is observed when comparing relapses from laparoscopic radical hysterectomy with vaginal radical hysterectomy (16% versus 7.69%). Nevertheless, in a recent randomised clinical trial [29], a comparison between open and laparoscopic radical trachelectomy was performed, and they found no differences in terms of DFS or OS.

Nevertheless, as we can observe in Figure 3, there a tendency for the experimental technique to be inferior in terms of DFS. Therefore, although strictly speaking we could not affirm or deny that FSS is inferior to the conventional technique for the treatment of initial cervical cancer, we cannot forget that there was a clear trend toward a decrease in the DFS when we applied the experimental technique, especially for tumour sizes of more than 2 cm. In other words, FSS may have a certain risk of recurrence and should only be offered if there is a significant desire to preserve fertility.

## 5. Strengths and Weaknesses

One of the main strengths of this study is that it offers a vision in terms of the status of the conservative treatment of cervical cancer in early stages over a wide period in the Spanish state. It also collects the results of tertiary hospitals in Spain dedicated to the treatment of gynaecological cancer. At the same time, this work allowed the creation of a network of centres dedicated to cervical cancer in our country from which numerous collaborations have arisen (Spain GOG cervical cancer taskforce). Another strength of this study is the considerable amount of statistical work carried out to be able to balance and extract reliable conclusions from the data. It is well known that a retrospective study has multiple biases that limit its clinical validity, but it is also true that well-performed propensity score matching makes the results obtained through this technique resemble a prospective study by equally balancing the arms of the study. In addition, with the introduction of noninferiority studies in daily clinical practice, the applicability of an experimental treatment with respect to the standard can be compared with great efficiency.

Regarding the weaknesses, one of the principal weak points of this study is its retrospective nature and the long duration of the analysis, nearly 14 years, during which some modifications were introduced progressively in the standard of care for the management of cervical cancer, including the introduction of sentinel node biopsy [30].

Another weakness of this study lies in the low number of cases resulting after balancing the study arms, which had an even greater effect, if possible, on the low frequency at which FSS was performed in our environment.

In conclusion, in this study, none of the analyses allowed us to demonstrate the inferiority of the experimental method compared to the standard method. Patients with tumours larger than 2 cm had a higher risk of recurrence regardless of the technique used. Therefore, according to our results, fertility preservation surgery in the initial stages of cervical cancer can be just as effective and safe as radical treatment.

## Figures and Tables

**Figure 1 jpm-12-01081-f001:**
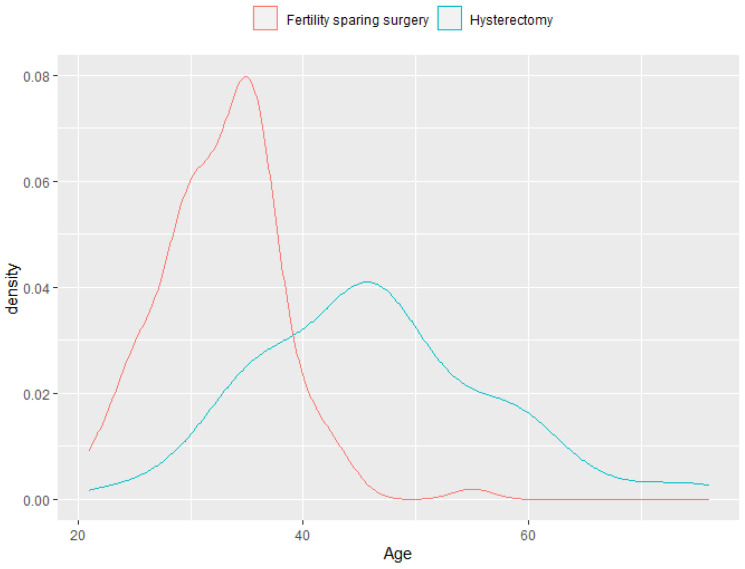
Distribution of the variable age at diagnosis in both treatment groups.

**Figure 2 jpm-12-01081-f002:**
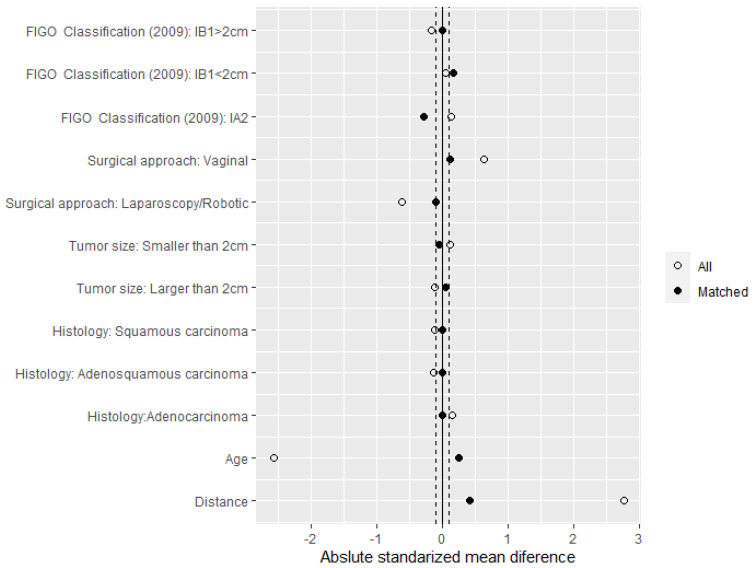
Love plot to visualise the goodness of the match.

**Figure 3 jpm-12-01081-f003:**
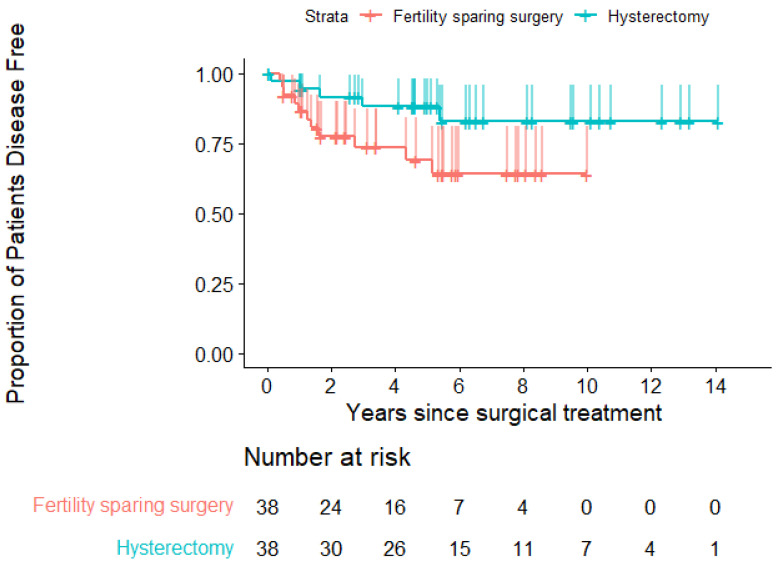
Kaplan–Meier disease-free survival curves. Tick marks indicate censored data. The vertical lines indicate the upper limits of the one-sided CI for noninferiority at 95%.

**Figure 4 jpm-12-01081-f004:**
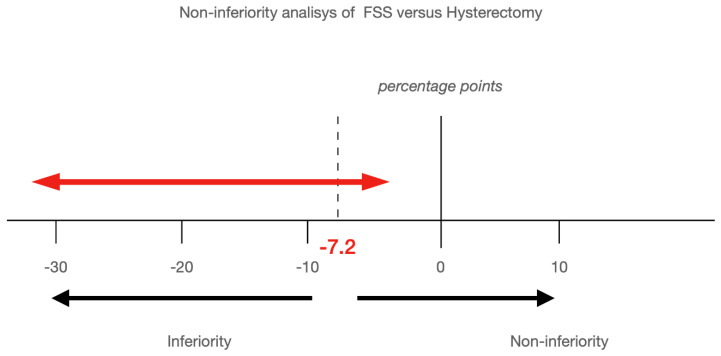
Noninferiority analysis between FSS and hysterectomy at 5 years.

**Table 1 jpm-12-01081-t001:** Clinicopathological characteristics of the unmatched series.

	Total *n* = 222	Fertility-Sparing Surgery *n* = 111	Radical Hysterectomy *n* = 111	*p* Value
Age	39.5 ± 10.6 years	46.2 ± 10 years	32.7 ± 5.3 years	0.001
Follow up	62.8 ± 45.1 months	65.9 ± 46.7 months	59.6 ± 46.4 months	0.299
FIGO Classification (2009)				0.294
IA1 plus LVI+	7	5 (4.5%)	2 (1.8%)	
IA2	25	15 (13.5%)	10 (9%)	
IB1 < 2 cm	133	67 (60.4%)	66 (59.4%)	
IB1 ≥ 2 cm	57	24 (21.6%)	33 (29.7%)	
Histology				0.4469
Adenocarcinoma	84 (37.8%)	45 (40.5%)	39 (35.1%)	
Squamous carcinoma	137 (61.7%)	66 (59.5%)	71 (64%)	
Adenosquamous carcinoma	1 (0.5%)	0	1 (0.9%)	
LVI				0.8444
Positive	30 (13.5%)	16 (14.4%)	14 (12.6%)	
Negative	192 (86.5%)	95 (85.6%)	97 (87.4%)	
Tumour size				0.3803
Less than 2 cm	155 (69.8%)	81 (73.0%)	74 (66.7%)	
Greater or equal than 2 cm	67 (30.2%)	30 (27.0%)	37 (33.3%)	
Nodal assessment				0.6971
PLD + SLN	165	80 (72.1%)	85 (76.6%)	
Only SLN	55	29 (26.1%)	26 (23.4%)	
Missing values	2	2 (1.8%)	0	
Intraoperative complications				0.48
Yes	9 (4.1%)	6 (5.4%)	3 (2.7%)	
No	213 (95.9%)	105 (94.6%)	108 (97.3%)	
Recurrence localizations				
Distance	2 (12.5%)	0 (0.0%)	2 (40.0%)	0.1532
Adnexal	2 (12.5%)	2 (18.2%)	0 (0.0%)	
Cervical	3 (18.8%)	3 (27.3%)	0 (0.0%)	
Lymph nodes	3 (18.8%)	2 (18.2%)	1 (20.0%)	
Local	6 (37.5%)	4 (36.4%)	2 (40.0%)	
Surgical approach				<0.001
Laparoscopy/Robotic	109 (49.1%)	39 (35.1%)	70 (63.1%)	
Laparotomy	4 (1.8%)	0 (0.0%)	4 (3.6%)	
Vaginal	109 (49.1%)	72 (64.9%)	37 (33.3%)	
Postoperative complications				0.001
Yes	7	0 (0%)	7 (6.3%)	
No	215	111 (100%)	104 (93.7%)	
Recurrence				0.1944
Yes	16 (7.2%)	11 (9.9%)	5 (4.5%)	
No	205 (92.8%)	100 (90.1%)	106 (95.5%)	

PLD: pelvic lymphadenectomy. SLN: sentinel lymph node. LVI: lymphovascular space invasion.

**Table 2 jpm-12-01081-t002:** Clinicopathological characteristics of the patients with recurrence.

			Patients with Recurrence		
	Patients with Recurrence (Recurrence/Total)	*p* Value	Fertility-Sparing Surgery *n* = 11	Radical Hysterectomy *n* = 5	*p* Value
Age	38.2 ± 12.3 years		31.18 ± 5.7 years	53.6 ± 7.5 years	0.0008
Follow-up	66.6 ± 46.8 months		77.94 ± 51.6 months	41.8 ± 21.2 months	0.067
Time to recurrence	23.11 ± 20.38 months		21.45 ± 19.29 months	26.75 ± 24.54 months	0.68
FIGO Classification (2009)		0.0343			0.294
IA1 y LVI +	0/7		0	0	
IA2	1/25 (4%)		1/11 (9.1%)	0	
IB1 < 2 cm	6/133 (4.5%)		5/11 (45.5%)	1/5 (20.0%)	
IB1 ≥ 2 cm	9/57 (15.8%)		5/11 (45.5%)	4/5 (80.0%)	
Histology		0.5669			1
Adenocarcinoma	8/84 (9.5%)		6/11 (54.5%)	2/5 (40.0%)	
Squamous carcinoma	8/137 (5.8%)		5/11 (45.5%)	3/5 (60.0%)	
Adenosquamous carcinoma	0/1		0	0	
LVI		0.3098			1
Positive	4/30 (13.3%)		3/11 (27.3%)	1/5 (20.0%)	
Negative	12/192 (6.2%)		8/11 (72.7%)	4/5 (80.0%)	
Tumour size					
Smaller than 2 cm	6/155 (3.9%)	0.0083	5/11 (45.5%)	1/5 (20.0%)	0.3803
Larger or equal than 2 cm	10/67 (14.9%)		6/11 (54.5%)	4/5 (80.0%)	
Intraoperative complications					
Yes	0/9		0	0	
No	16/212 (8.3%)		11/11 (100%)	5/5 (100%)	
Recurrence localizations					
Distance	2		0 (0.0%)	2 (40.0%)	0.1532
Adnexal	2		2 (18.2%)	0 (0.0%)	
Cervical	3		3 (27.3%)	0 (0.0%)	
Lymph nodes	3		2 (18.2%)	1 (20.0%)	
Local	6		4 (36.4%)	2 (40.0%)	
Surgical approach					
Laparoscopy/Robotic	10/109 (9.2%)		6 (54.5%)	4 (80.0%)	0.5879
Laparotomy	0		0 (0.0%)	0 (0.0%)	
Vaginal	6/109 (5.5%)		5 (45.5%)	1 (20.0%)	
Postoperative complications					
Yes	0		0	0	
No	16/206		11(100%)	5(100%)	

LVI: lymphovascular space invasion.

**Table 3 jpm-12-01081-t003:** Clinicopathological characteristics of the matched series.

	Matched Subsample *n* = 76	Fertility-Sparing Surgery *n* = 38	Radical Hysterectomy *n* = 38	*p*-Value
Age	35.9 ± 7.1 years	34.2 ± 5.5 years	37.7 ± 8.1 years	0.189
Follow-up	65.7 ± 43.1 months	61.3 ± 40.5 months	70.1 ± 45.6 months	0.3749
FIGO Classification (2009) IA1 and LVI+	0	0	0	0.5435
IA2	9 (11.8%)	3 (7.9%)	6 (15.8%)	
IB1 < 2 cm	41 (53.9%)	22 (57.9%)	19 (50.0%)	
IB1 > 2 cm	26 (34.2%)	13 (34.2%)	13 (34.2%)	
Histology				
Adenocarcinoma	28 (36.8%)	14 (36.8%)	14 (36.8%)	1
Squamous carcinoma	48 (63.2%)	24 (63.2%)	24 (63.2%)	
Adenosquamous carcinoma	0	0	0	
LVI				
Positive	14 (18.4%)	8 (21.1%)	6 (15.8%)	0.7673
Negative	62 (81.6%)	30 (78.9%)	32 (84.2%)	
Tumour size				
Smaller than 2 cm	45 (59.2%)	22 (57.9%)	23 (60.5%)	1
Larger than 2 cm	31 (40.8%)	16 (42.1%)	15 (39.5%)	
Intraoperative complications				
Yes	4 (5.3%)	3 (7.9%)	1 (2.6%)	0.6075
No	72 (94.7%)	35 (92.1%)	37 (97.4%)	
Recurrence localizations				
Distance	2 (12.5%)	0 (0.0%)	2 (40.0%)	0.1532
Adnexal	2 (12.5%)	2 (18.2%)	0 (0.0%)	
Cervical	3 (18.8%)	3 (27.3%)	0 (0.0%)	
Lymph nodes	3 (18.8%)	2 (18.2%)	1 (20.0%)	
Local	6 (37.5%)	4 (36.4%)	2 (40.0%)	
Surgical approach				
Laparoscopy/robotic	48 (63.2%)	23 (60.5%)	25 (65.8%)	0.812
Laparotomy	0	0	0	
Vaginal	28 (36.8%)	15 (39.5%)	13 (34.2%)	
Recurrence				
Yes	16 (21.1%)	11 (28.9%)	5 (13.2%)	0.1595
No	60 (78.9%)	27 (71.1%)	33 (86.8%)	

**Table 4 jpm-12-01081-t004:** Factors associated with disease-free survival (balanced data).

	Univariate Analysis		Multivariate Analysis	
Factor	Hazard Ratio (95% CI)	*p*-Value	Hazard Ratio (95% CI)	*p*-Value
Age	1.06 (0.98–1.13)	0.101	1.05 (0.99–1.12)	0.06
Tumour size larger than 2 cm vs. tumour size smaller than 2 cm	2.091 (1.02–4.28)	0.04	1.99 (0.94–4.19)	0.07
Squamous carcinoma vs. adenocarcinoma	0.59 (0.22–1.57)	0.288	0.59 (0.20–1.76)	0.34
Vaginal surgical approach vs. laparoscopy/robotic	0.82 (0.29–2.27)	0.702	0.91 (0.31–2.66)	0.87
LVI-positive vs. LVI-negative	2.101 (0.67–6.54)	0.2	1.83 (0.47–7.01)	0.37

**Table 5 jpm-12-01081-t005:** Association between different factors and the disease-free survival time in each surgical group expressed in terms of hazard ratios and their 95% CIs.

	Fertility-Sparing Surgery *n* = 38	Hysterectomy *n* = 38	HR (Fertility-Sparing Surgery vs. Hysterectomy) (95% CI)
Matched subsample:	11/38 (28.9%)	5/38 (13.2%)	2.5 (0.89; 7.41)
Tumour size: smaller than 2 cm	5/22 (22.72%)	1/23 (4.35%)	5.90 (0.69; 50.63)
Tumour size: greater than 2 cm	6/16 (37.5%)	4/15 (26.67%)	1.71 (0.48; 6.11)
Histology: adenocarcinoma	6/14 (42.86%)	2/14 (14.29%)	3.84 (0.77; 19.20)
Histology: squamous carcinoma	5/24 (20.83%)	3/24 (12.5%)	1.87 (0.44; 7.85)
Surgical approach: laparoscopic/robotic	6/23 (26.09%)	4/25 (16%)	1.85 (0.52; 6.61)
Surgical approach: vaginal	5/15 (33.33%)	1/13 (7.69%)	6.33 (0.73; 54.99)
Figo Classification (2009): IB1 < 2 cm	5/22 (22.73%)	1/19 (5.26%)	5.35 (0.62; 45.99)
Figo Classification (2009): IB1 > 2 cm	5/13 (38.46%)	4/13 (30.77%)	1.50 (0.40; 5.67)
LVI: Negative	8/30 (26.67%)	4/32 (12.5%)	2.55 (0.77; 8.53)
LVI: Positive	3/8 (37.5%)	1/6 (16.67%)	2.74 (0.28; 26.68)

* The second and third columns indicate the number of recurrences in each case divided by the total number of cases in the group.

**Table 6 jpm-12-01081-t006:** Differences in disease-free survival rates at 2.5 and 5 years after surgery.

Years from Surgery	Disease-Free Recurrence Rate (Rate (95% CI))		Difference (95% CI)
	Fertility-Sparing Surgery	Hysterectomy	
2.5 years	77.46 (64.79–92.60)	91.40 (82.53–100)	−13.94 (−24.84, −3.03)
5 years	68.99 (54.22–87.77)	88.01 (77.59–99.83)	−19.02 (−32.08, −5.96)

## Data Availability

The data that support the findings of this study are available from the corresponding author upon reasonable request.

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
