# Peer review of "Fertility-Sparing Surgery versus Radical Hysterectomy in Early Cervical Cancer: A Propensity Score Matching Analysis and Noninferiority Study"

_jpm, 2022, doi:10.3390/jpm12071081_

Round 1
Reviewer 1 Report
This is a well composed retrospective study with particular importance for the Spanish Gynecologic Oncology cervical cancer working Group. The author`s aim was to analyze the oncological outcomes of Fertility-sparing surgery (FSS) compared to a balanced group of standard radical hysterectomy (RH) in patients with early cervical cancer (ECC).
The authors conclude that in their material FSS offers excellent disease-free and overall survival in women with ECC with fertility desire and is not inferior compared to RH. Irrespective of statistical significance terms, it is evident that recurrence rates were unfavorable for the FSS group (28.9% in the FSS group vs. 13.2% in RH group). Furthermore, the rate of disease-free survival at 5 years was 68.99% in the FSS group and 88.01% in the RH group. Among the 16 patients with disease recurrence, 11 patients belonged to the FSS group (recurrence rate 9.9%) and 5 patients belonged to the conventional surgery group (recurrence rate 4.5%). These figures could challenge the non-inferiority conclusion of the study.
Additionally, during a 14 year study span, surgical procedures differ as surgical experience accrues, new findings evolve from studies (e.g. sentinel node procedures) and surgical teams subspecialize. There is a question regarding the consistency/ homogeneity of the procedures; the authors are knowledgeable however this represents an intrinsic confounding factor which is difficult to mitigate.
Finally, despite outside of the scopes of the study, a comparison of psychosexual scores between the two groups (e.g. in the form of a questionnaire) would be meaningful. Additionally, many readers would be interested in conception rates as well as the take-home-baby rate for the FSS group.
Reviewer 2 Report
Abstract - line 22 - "Fertility-sparing therapy" is not necessarily to be with a capital letter. Line 24 - the study Design, Setting and patients should be addressed in the methods section, but not in the Objectives part of the abstract. Please structure your abstract appropriately. It must be improved, especially the Conclusion.
Introduction. In line 47 a full stop is missing before "Novak". In general, the introduction part is too narrow and does not give a clear study rational. Please provide some epidemiological data about cervical cancer among your population (the population studies in this research). Prevalence of specific surgical interventions (radical hysterectomy, trachelectomy) with its mortality/survival rates would also make the introduction part more attractive to a potential reader.
The metods section is detailed enough. However, it could benefit if better structured with subheadings (example - study design; study subjects, variables, statistical analysis, ethical approval).
The study results are described in a great details and supported by tables and figures. However, the stat methods applied (lines 248-251) could sound better in the methods section (statistical analysis subsection).
The discussion part and the conclusion are clear.
